# Factors associated with insufficient weight gain among Mexican pregnant women with HIV infection receiving antiretroviral therapy

Estela Godínez[1], Mayra Chávez-Courtois[1]*, Ricardo Figueroa[2], Rosa María Morales[1], Cristina Ramírez[1], Maricruz Tolentino[1]

1 Departamento de Nutrición y Bioprogramación, Instituto Nacional de Perinatología, Ciudad de México, México, 2 Departamento de Infectología, Instituto Nacional de Perinatología, Ciudad de México, México

☯ These authors contributed equally to this work.
‡ These authors also contributed equally to this work.
* courml@yahoo.com.mx

## Abstract

### Objective

We identified clinical, dietary, and socioeconomic factors associated with insufficient gestational weight gain among Mexican pregnant women with human immunodeficiency virus (HIV) infection.

### Methods

This was a cross-sectional study involving 112 pregnant women with HIV infection receiving antiretroviral therapy (ART). Data including viral load, complete blood analysis, and CD4 counts were extracted from medical records. An inquiry form was used to collect data on socioeconomic status and frequency of food intake. Pre-gestational weight was calculated based on pregnancy weight to obtain the body mass index (BMI) and weight gain for gestational age according the US Institute of Medicine. Of the study population, 68.7% were in consensual union, 31.3% were single, and 33.9% belonged to the two lowest socioeconomic strata. The median age and CD4 count were 27 (interquartile range [IQR]: 23–32) years and 418 (IQR: 267–591), respectively. The adequacy of energy was 91.8% (IQR: 74.1–117.7). The median energy intake from protein was 13.5% (IQR: 12.2–14.9) and from lipids, 35.5% (IQR: 31.1–40.3). Pregnant women with gastrointestinal symptoms and CD4 count <350 were seven times more likely to have folate deficiency (odds ratio [OR] 7.8, 95% confidence interval [CI] 1.6–38.1; p = 0.009) and six times more likely to have poor zinc intake (OR 6.7, 95% CI 1.3–36.8; p = 0.014). In all, 42.9% of the pregnant women consumed iron and folic acid supplements and 54.4% consumed multivitamin supplements. Moreover, 45.5% had a normal pre-gestational BMI, 41.1% were classified overweight, and 13.4% had obesity, whereas 62.5% showed insufficient gestational weight gain, and 18.8% experienced weight loss. The variables associated with insufficient weight gain were consensual union (OR 5.3, 95% CI 1.9–15.0; p = 0.002) and belonging to the lowest socioeconomic stratum (E) (OR 3.1, 95% CI 1.0–9.2; p = 0.046).

**Data Availability Statement:** All relevant data are within the manuscript and its Supporting Information file

**Funding:** Unfunded study.

**Competing interests:** The authors have declared that no competing interests exist.

## Conclusion

Dietary strategies to improve gestational weight gain for Mexican women with HIV infection receiving ART must consider clinical and socioeconomic factors.

## Introduction

Pregnancy increases energy and nutrient requirements which are necessary to provide a healthy intrauterine environment, allowing the optimal development of the fetus [1]. However, patients with human immunodeficiency virus (HIV) receiving antiretroviral therapy (ART) can see an increase in basal energy expenditure of up to 20% [2]. Likewise, food intake can be affected by HIV infection, the existence of opportunistic infections and the secondary effects of treatment such as nausea, vomiting, diarrhea, and alterations in taste and smell [3]. Although it is well-known that this is unlikely to affect gestational weight gain in women who are HIV seropositive living in developed countries, more research is needed among HIV seropositive pregnant women in developing countries who often experience poverty [2] because a large proportion of them live in economic detriment, isolation and social stigmatization resulting in food insecurity[4]. Accordingly, it is expected that typical diet of this population is influenced by food security and food distribution as well as by the educational level and economic status [5]. Thus, it has been proposed to understand the bidirectional relationship between food insecurity and HIV progression, at different societal levels (community, family and individual). Because of this, the family structure and social support become fundamental to food insecurity within the domestic unit [6]. The dynamics of the domestic unit are based on gender and age, cooperative relationships, and the power exchange and conflict that are continuously being established around the division of labor and decision making [7]. These dynamics can affect the diet of family members both by gender and age [8]. As a result, the nutritional status of HIV seropositive pregnant women is affected not only by the metabolic and physiological changes of pregnancy [9,10], but also by other socioeconomic determinants and family dynamics that impact access to an adequate diet. All the above-mentioned factors can result in insufficient gestational weight gain which may lead to an increase in maternal and neonatal morbidities in both the short- and long-term [11,12].

In cohort studies of HIV seropositive pregnant women who started ART, it was found that those who had a weekly gestational weight gain of less than 100 grams displayed a higher risk of preterm birth and adverse pregnancy-neonatal outcomes such as low birth weight suggesting that the use of ART cannot solely reduce the risk of these outcomes, because of pre-existing poor nutritional status external to ART [12,13]. Moreover, pregnant women without HIV insufficient weight gain are also associated with adverse neonatal outcomes, so it will be not be possible for ART alone to resolve problems of low gestational weight gain. [14–16]. Because the effect on nutritional status indicators such BMI and hemoglobin concentrations are not sustained and alterations in these indicators may occur despite ART administration [17], there are other factors that also could affect the nutritional status of this vulnerable population. In this study, we aimed to identify the clinical, dietetic, and socioeconomic factors that are associated with insufficient gestational weight gain among HIV seropositive pregnant women receiving ART.

## Materials and methods

This research was approved by the institutional committees of ethics and research; National Institute of Perinatology, Mexico, Number 212250–49561. All participant's gave consent to participate in the study.

From January 2013 to January 2017, a cross-sectional study was performed at the National Institute of Perinatology, Mexico. This is a third-level care institution supporting women with high-risk pregnancies, mainly living in Mexico City. The study included 112 HIV seropositive pregnant women who were in their second or third trimesters of pregnancy and had no other identified comorbidity. The patients were recruited through their prenatal care consultation at the Institute, verbal and written informed consent was requested to women and guardians of minor's. The participants formed a non-random convenience sample.

The survey referred to as the "10 x 6 Rule" was used to collect information on patients' socioeconomic status [18]. Patients were also asked about the presence of gastrointestinal symptoms such as nausea, vomiting, diarrhea and swallowing difficulty, during ART application and as soon as they knew they were pregnant. To qualify as chronic symptoms, the symptoms should have occurred at least twice a week. To calculate the energy and nutrient consumption, we used the semiquantitative questionnaire evaluating the food intake frequency in the last three months [19]. The percentages of energy adequacy were calculated in calories in reference to the total energy expenditure obtained from the sum of Basal Energy Expenditure (BEE) calculated using the Harris Benedict formula [20], plus the thermogenic effect of food (10% BEE), a sedentary activity (20% BEE), an increase due to the presence of HIV (10% BEE for CD4$\geq$200 and 20% BEE for CD4 <200 [21]), and the expense for pregnancy (350 calories for the 2nd quarter and 462 for the 3rd) [22]. Supplements and/or multivitamins intake were also recorded. The daily nutrient intake was assessed with respect to the recommended daily intake for the Mexican population [23] and insufficient consumption was considered to be present when the percentage of self-reported intake was less than 90% of what was recommended.

Prepregnancy weight was calculated according to a model that estimates the pre-gestational weight from the first visit weight during pregnancy by considering the height, gestational age, current weight, parity, and age using a software (https://www.pbrc.edu/research-and-faculty/calculators/prepregnancy/) [24]. To achieve these body measurements, patients wore a clinical gown and were weighed using a calibrated scale (SECA, professional model 703 Seca North America, Hanover, MD, USA). Height was measured using an electronic stadiometer (model 242 Seca North America, with the Lohman technique) [25]. With the current weight and height attained, appropriate gestational weight gain was classified for the corresponding week of gestation according to the pre-pregnancy body mass index (BMI) using the criteria of the United States Institute of Medicine [26].

Through the review of the clinical records, we collected data regarding the women's schooling, marital status, gynecological antecedents, current ART, time of use of ART, hemoglobin concentrations, CD4 cell count and viral load.

Descriptive statistics were calculated. To evaluate association between variables, a chi-squared test and odds ratios (OR) was calculated with 95% confidence intervals (CI). Correlation between variables was evaluated using the Spearman index. Finally logistic regression analysis was performed to evaluate the association of socioeconomic, dietetic, and clinical variables with insufficient weight gain.

## Results

The socioeconomic and obstetric features by age in this study are shown in Table 1. Most (73.2%) women were aged between 19 and 24 years and were living in consensual union (68.7%) with their current HIV-negative partner (61.6%). Approximately 76.8% of the women were home-based and belonged to the lowest socioeconomic strata "E" (33.9%) and "D" (37.5%). Indeed, 34.8% had junior high school education. Regarding their reproductive

**Table 1. Socioeconomic and obstetric features of HIV seropositive pregnant women by age.**

| Variable | Total n = 112 (100%) | ≤18 years n = 10 (8.9%) | 19–34 years n = 82 (73.2%) | >35 years n = 20 (17.9%) |
|---|---|---|---|---|
| Marital Status* | | | | |
| Single | 35 (31.3) | 5 (4.5) | 23 (20.5) | 7 (6.3) |
| Free union | 55 (49.1) | 4 (3.6) | 47 (42.0) | 4 (3.6) |
| Married | 22 (19.6) | 1 (0.9) | 12 (10.7) | 9 (8) |
| Occupation* | | | | |
| Home | 86 (76.8) | 6 (5.4) | 67 (59.8) | 13 (11.6) |
| Student | 4 (3.6) | 3 (2.7) | 1 (0.9) | 0 (0) |
| Remunerated job | 22 (19.6) | 1 (0.9) | 14 (12.5) | 7 (6.3) |
| Socioeconomic strata | | | | |
| E | 38 (33.9) | 2 (1.8) | 30 (26.8) | 6 (5.4) |
| D | 42 (37.5) | 4 (3.6) | 31 (27.7) | 7 (6.3) |
| D+ | 24 (21.4) | 3 (2.7) | 17 (15.2) | 4 (3.6) |
| C | 6 (5.4) | 1 (0.9) | 3 (2.7) | 2 (1.8) |
| C+ | 1 (0.9) | 0 (0) | 0 (0) | 1 (0.9) |
| AB | 1 (0.9) | 0 (0) | 1 (0.9) | 0 (0) |
| Educational attainment | | | | |
| Without studies | 5 (4.5) | 0 (0) | 4 (3.6) | 1 (0.9) |
| Primary Education | 29 (25.9) | 1 (0.9) | 22 (19.6) | 6 (5.4) |
| Junior High school | 39 (34.8) | 4 (3.6) | 29 (25.9) | 6 (5.4) |
| High school /Technical | 27 (24.1) | 5 (4.5) | 18 (16.1) | 4 (3.6) |
| Professional | 12 (10.7) | 0 (0) | 9 (8.0) | 3 (2.7) |
| Current partner HIV+ | | | | |
| Yes | 41 (36.6) | 2 (1.8) | 34 (30.4) | 5 (4.5) |
| No | 69 (61.6) | 7 (6.3) | 47 (42) | 15 (13.4) |
| Unknow | 2 (1.8) | 1 (0.9) | 1 (0.9) | 0 (0) |
| Gravidity* | | | | |
| 1 | 25 (22.3) | 5 (4.5) | 16 (14.3) | 4 (3.6) |
| 2–3 | 64 (57.1) | 5 (4.5) | 51 (45.5) | 8 (7.1) |
| ≥4 | 23 (20.5) | 0 (0) | 15 (13.4) | 8 (7.1) |
| Abortion | | | | |
| 0 | 90 (80.4) | 10 (8.9) | 66 (58.9) | 14 (12.5) |
| 1–2 | 21 (18.8) | 0 (0) | 15 (13.4) | 6 (5.4) |
| 3–4 | 1 (0.9) | 0 (0) | 1 (0.9) | 0 (0) |
| Parity* | | | | |
| 0 | 52 (46.4) | 9 (8.0) | 37 (33.0) | 6 (5.4) |
| 1–2 | 47 (42.0) | 1 (0.9) | 36 (32.1) | 10 (8.9) |
| ≥3 | 13 (11.6) | 0 (0) | 9 (8.0) | 4 (3.6) |

Chi$^2$ Test was performed.

*p < .05

Values in table are expressed as frequency and % and chi-squared test was performed to compare groups

antecedents, most were already in their second or third pregnancy (57.1%) and 80.4% had not experienced any gestational losses.

The median CD4 count was 418.5 (interquartile range (IQR) 267.2–591.5). Table 2 shows the clinical features of the women according to their CD4 count. It was observed that those with CD4 count of <350 had been on ART for a shorter duration [8 months (IQR: 2.0–44.7)

*vs* 33 months (IQR: 6.0–71.2); p = 0.005] and had lower concentrations of hemoglobin [11.6g/L±1.6 *vs* 12.4g/L±1.1; p = 0.018] compared to those with a CD4 count of ≥350.

Regarding diet, there was a higher proportion of women with insufficient energy intake in those who were at an advanced stage of this disease (76.4%) compared to those who were in the early and intermediate stages (23.6%) (Chi$^2$ = 6.0; p = 0.014).

The percentage of adequacy of consumption of energy (r = 0.21), iron (r = 0 .28), zinc (r = 0.29), protein (r = 0.28), total monounsaturated fats (r = 0.26), and fiber (r = 0.22) was correlated with the socioeconomic level (p<0.05). In addition, a higher proportion of women (59.6%) with low zinc and vitamin C intake belonged to the lowest socioeconomic strata (p = 0.04: Chi$^2$ = 4.2; p = 0.06 Chi$^2$ = 3.5), and 42.9% of women reported that they consumed iron and folic acid supplements, and 54.4% reported of consuming multivitamin supplements.

Most of the women reported at least one gastrointestinal symptom due to ART (52.7%), the most frequent being nausea (42%), vomiting (27.7%) and diarrhea (21.4%). On the other hand, a large proportion of women (75.9%) manifested some gastrointestinal symptoms due to pregnancy, the most common being nausea (52.7%) and vomiting (49.1%) and the least frequently being diarrhea (18.8%). Table 3 shows that women who presented at least one gastrointestinal symptom due to pregnancy were about three times more likely to have an unsatisfactory folic acid (OR 2.9, 95% CI 1.2–7.1; p = 0.019), vitamin c (OR 2.5, 95% CI 1.0–6.6; p = 0.048) and zinc (OR 3.3, 95% CI 1.2–8.9, p = 0.014) intake compared to those that did not present symptoms. This association was stronger in those with CD4 <350 for folic acid and zinc intakes.

Most of the women presented a normal pre-pregnancy BMI (45.5%), and 41.1% were overweight whereas 13.4% were obese. However, 62.5% showed an insufficient weight gain and 19% had weight loss at the time of the evaluation. Moreover, 13.4% had a weight gain according to weight gain recommendations and only 5.4% had excessive weight gain. Pre-pregnancy BMI and weight gain did not correlate with the CD4 count (p>0.05).

Table 4 shows variables that correlated with insufficient gestational weight gain and weight loss in those who had a CD4 count of <350. These correlations were: marital status (r = 0.34; p = 0.028), hemoglobin concentrations (r = -0.30, p = 0.053), the presence of nausea and vomiting (r = 0.33; p = 0.03) and protein intake (r = -0.38; p = 0.014). On the contrary, in those with CD4≥350, the correlated data were: marital status (r = 0.25; p = 0.032), socioeconomic status (r = -0.27; p = 0.023) and multivitamin use (r = -0.24; p = 0.039). On the other hand, weight loss correlated with protein intake (r = -0.25; p = 0.007 and in those with CD4 count <350, there was a tendency for insufficient energy intake (r = -0.30; p = 0.058).

Table 5 shows the variables that were significantly associated with insufficient gestational weight gain or weigh loss for women of this study are: consensual union (in free union or married) and belonging to the lowest socioeconomic level (E).

## Discussion

Many factors related to weight gain during pregnancy, have been evaluated in several studies [27–33]. However, few have been completed in HIV seropositive women [33]. The most relevant literature focused on a rural African population where HIV prevalence is much higher than in other countries; in addition, in this region, pattern of ART provision is different to that in Mexico [34]. In Mexico, access to ART occurs mainly in urban populations where the present study was performed.

Although it has been described that BMI is associated with CD4 count [3], in this study, no association was observed between the pregestational BMI and the CD4 count. There was also no relationship between the CD4 count and weight gain, unlike the study by Villamor et al.,

**Table 2. Clinical and dietetic features of HIV seropositive pregnant women according to their CD4 count.**

| Features | Total (n = 112) | | | CD4 <350(n = 41) | CD4 ≥350(n = 71) |
|---|---|---|---|---|---|
| | Median/(IQR) | Interval | | Median/(IQR) | Median/(IQR) |
| Clinic | | | | | |
| GA (weeks)[a] | 29.0 (23.2–34.0) | 12–38 | | 33.0 (25.5–35.5) | 27.0 (19.0–32.0) |
| Age (years) | 27.0 (23.0–31.7) | 16–45 | | 27.0 (23.0–30.0) | 28.0 (23.0–33.0) |
| Weight PG (kg) | 60.7 (54.8–69.3) | 41.7–100.8 | | 61.0 (54.4–68.7) | 60.0 (54.8–69.5) |
| Height (cm)* | 154.3 ± 6.3 | 132–167 | | 153.2 ± 5.8 | 154.7 ± 6.7 |
| BMI PG (m²/kg) | 25.8 (23.2–28.0) | 18.7–39.4 | | 25.7 (23.1–28.6) | 25.9 (23.2–27.9) |
| $t$ Tx. ARV (months)[b] | 24.0 (3.7–61.5) | 0–294 | | 8 (2.0–44.7) | 33 (6.0–71.2) |
| Hemoglobin (g/L)*[c] | 12.1 ± 1.3 | 6.6–14.8 | | 11.6 ± 1.6 | 12.4 ± 1.1 |
| Dietetic | | | | | |
| Energy (%A[1]) | 91.8 (74.1–117.7) | 33.5–197.3 | | 88.3 (71.0–119.0) | 94.2 (78.7–117.0) |
| Carbohydrates (%[2]) | 51.9 (47.0–57.0) | 29.0–68.0 | | 51.5 (47.8–58.0) | 52.1 (46.8–56.2) |
| Protein (%[2])[d] | 13.5 (12.2–14.9) | 7.8–18.8 | | 12.8 (11.8–14.6) | 13.6 (12.5–15.2) |
| Fat (%[2]) | 35.5 (31.0–40.3) | 17.7–52.6 | | 37.3 (30.6–40.2) | 35.0 (31.1–40.6) |
| Folic Acid (%A[1])[e] | 64.1 (35.6–131.5) | 15.2–309.2 | | 52.6 (29.4–91.8) | 72.3 (40.5–152.2) |
| Iron (%A[1]) | 40.2 (31.1–52.3) | 12.5–87.3 | | 40.2 (30.8–49.6) | 40.1 (31.7–53.9) |
| Zinc (%A[1])[f] | 98.6 (64.7–139.6) | 24.2–298.4 | | 92.6 (53.8–122.1) | 112.7 (71.0–149.4) |
| Fiber (g) | 25.7 (20.3–33.7) | 11.5–62.1 | | 25.5 (19.8–31.1) | 26.2 (20.3–35.3) |

IQR = interquartile range; GA = gestational age; PG = pregestational; BMI = body mass index; $t$ Tx. ARV = time of use of antiretroviral therapy.

[1]Percentage of adequacy with relation to recommended or suggested daily intake for Mexican population (18)

[2]Percentage of the total energy value.

U de Mann Whitney test was performed

*Median ± SD: T student test was performed

[a]p = 0.000

[b]p = 0.005

[c]p = 0.018

[d]p = 0.076

[e]p = 0.056

[f]p = 0.052

[33] involving HIV seropositive women without ART and low CD4 counts, where CD4 count <200 was associated with lower weight gain. It is likely that in our study, the CD4 count was not related to the pre-pregnancy BMI nor affected the weight gain because only 15.2% had CD4 count of <200. Those with CD4 count of <350 were more likely to have lower hemoglobin levels and this is an association already reported and is related to the progression of the disease [35].

Although in other studies conducted in pregnant women without HIV infection [29–31], parity was positively related to insufficient weight gain; no such observation was noted in this study.

Regarding marital status, it was observed that pregnant women having consensual union were five times more likely to experience insufficient weight gain (OR 5.3, 95% CI 1.9–15.0; p = 0.002) compared to single women. This is consistent with previous findings by Changamire et al., in which unmarried pregnant women in Tanzania without HIV infection gained 97 g more per month than the free-union living individuals [32]. These results suggest that living with a partner is a factor associated insufficient weight gain. This finding may be related to food family dynamics where women are continuing serving other family members and are

**Table 3. Correlation of gastrointestinal symptoms in pregnancy and underconsumption.**

| Total (n = 112) | GI symptoms in pregnancy | | OR (CI 95%) | p |
|---|---|---|---|---|
| Under consumption | No f (%) | Yes f (%) | | |
| Vitamin D | | | | |
| No | 18 (16.1) | 40 (35.7) | 2.2 (0.9–5.6) | .073 |
| Yes | 9 (8.0) | 45 (40.2) | | |
| Vitamin C | | | | |
| No | 20 (17.9) | 45 (40.2) | 2.5 (1.0–6.6) | .048 |
| Yes | 7 (6.3) | 39 (34.8) | | |
| Folic Acid | | | | |
| No | 14 (12.5) | 23 (20.5) | 2.9 (1.2–7.1) | .019 |
| Yes | 13 (11.6) | 62 (55.4) | | |
| Zinc | | | | |
| No | 21 (18.8) | 44 (39.3) | 3.3 (1.2–8.9) | .014 |
| Yes | 6 (5.4) | 41 (36.6) | | |
| CD4 < 350 (n = 41) | GI symptoms in pregnancy | | OR (CI 95%) | p |
| Under consumption | No f (%) | Yes f (%) | | |
| Vitamin D | | | | |
| No | 8 (19.5) | 12 (29.3) | 4.0 (0.9–18.2) | .060 |
| Yes | 3 (7.3) | 18 (43.9) | | |
| Vitamin C | | | | |
| No | 8 (19.5) | 13 (31.7) | 3.5 (0.8–15.8) | .090 |
| Yes | 4 (7.3) | 17 (41.5) | | |
| Folic Acid | | | | |
| No | 6 (14.6) | 4 (9.8) | 7.8 (1.6–38.1) | .009 |
| Yes | 5 (12.2) | 26 (63.4) | | |
| Zinc | | | | |
| No | 9 (22.0) | 12 (29.3) | 6.7 (1.3–36.8) | .014 |
| Yes | 2 (4.9) | 18 (43.9) | | |
| CD4 ≥350 (n = 71) | GI symptoms in pregnancy | | OR (CI 95%) | p |
| Under consumption | No f (%) | Yes f (%) | | |
| Vitamin D | | | | |
| No | 10 (14.1) | 28 (39.4) | 1.6 (0.5–5.0) | .411 |
| Yes | 6 (8.5) | 27 (38.0) | | |
| Vitamin C | | | | |
| No | 12 (16.9) | 32 (45.1) | 2.2 (0.6–7.5) | .212 |
| Yes | 4 (5.6) | 23 (32.4) | | |
| Folic Acid | | | | |
| No | 8 (11.3) | 19 (26.8) | 1.9 (0.6–5.8) | .267 |
| Yes | 7 (11.3) | 36 (50.7) | | |
| Zinc | | | | |
| No | 12 (16.9) | 32 (45.1) | 2.1 (0.6–7.5) | .212 |
| Yes | 4 (5.6) | 23 (32.4) | | |

GI = gastrointestinal; OR = Odds ratio; CI = confidence interval

responsible for family member's quality of health before considering their own health status. A study investigating family habits found that women were engaged in household activities in both rural and urban families and were responsible for food preparation, but food distribution

**Table 4. Correlation between socioeconomic, clinical, and dietary factors and insufficient weight gain and weight loss in HIV seropositive pregnant women.**

| Variable Insufficient weight gain and weight loss (n = 91) | Total (n = 112) | | CD4<350 (n = 41) | | CD4≥350 (n = 71) | |
|---|---|---|---|---|---|---|
| | r | P | R | p | R | p |
| Marital Status (Single, FU, Married) | .286 | .002 | .343 | .028 | .255 | .032 |
| Socioeconomic strata[1] | -.240 | .011 | -.201 | .208 | -.270 | .023 |
| Educational attainment[2] | -.123 | .197 | -.134 | .405 | -.142 | .236 |
| Height (cm) | -.200 | .034 | -.222 | .164 | -.205 | .087 |
| BMI Pregestational ($m^2$/kg) | -.221 | .019 | -.453 | .003 | -.081 | .503 |
| Parity (number of living children) | .051 | .597 | .072 | .657 | .044 | .716 |
| Hemoglobin (g/L) | -.160 | .094 | -.304 | .053 | -.064 | .600 |
| Nausea due to ARV[3] | .038 | .693 | .312 | .047 | .125 | .298 |
| Vomiting due to ARV[3] | .042 | .664 | .181 | .256 | -.043 | .723 |
| Nausea and/or vomiting due to ARV[3] | .063 | .507 | .334 | .033 | -.097 | .420 |
| Supplementation with MV[4] | -.164 | .085 | -.036 | .823 | -.245 | .039 |
| Protein intake (% of TEV) | -.119 | .211 | -.380 | .014 | -.046 | .702 |
| Weight loss (n = 21) | | | | | | |
| Insufficient intake of energy[3] | -.106 | .267 | -.299 | .058 | .003 | .978 |
| Protein intake (% of TEV) | -.255 | .007 | -.274 | .083 | .219 | .066 |
| Supplementation with MV[4] | -.158 | .096 | -.186 | .244 | .119 | .322 |

r = correlation coefficient; FU = free union; ARV = antiretroviral; TEV = Total energy value; PG = pregestational; Fe = Iron; FA = Folic Acid; MV = Multivitamin.

[1]E,D,D+,C,C+,AB

[2]without studies, primary education, junior high school, high school/technical, professional

[3]yes/no. Spearman correlations was performed.

depended on the decision of the bread winner, which was usually the man. This family dynamic reinforces gender inequality in the distribution of food by endorsing remunerated labor activities as reward in purchasing and receiving higher quality and portions of food [36]. Neither the health condition nor the reproductive status is considered when granting privileges in the alimentary family dynamics [36]. The opposite trend was noted in HIV seropositive single pregnant women, who had a lower prevalence of insufficient weight gain than those who were in consensual union (19.6 vs 61.6%, $Chi^2$ = 11.3, p = 0.001). In this group, it could be that the relationships within the domestic unit that comprise the gender, age and place within the family—daughters into the family—[8], situate them as part of the family members that have to be cared for especially because of their health and reproductive status, privileging them in the decision of the type of family life strategies [36]. This explains that in the HIV seropositive pregnant women without a partner, the distribution, quantity and quality of food within

**Table 5. Variables that favor insufficient weight gain in HIV seropositive pregnant women.**

| Variable | OR | CI 95% | p |
|---|---|---|---|
| Consensual union (free union or married) | 5.33 | 1.89–15.03 | .002 |
| Very low socioeconomic strata (E) | 3.06 | 1.02–9.21 | .046 |
| Protein intake (<12% TEV) | 0.65 | 0.19–2.26 | .502 |
| CD4 cells count (<350 cells/$mm^3$) | 1.03 | 0.35–3.03 | .952 |

%TEV = Percentage of the total energy value. Logistic regression analysis was performed. Cox and Snell $r^2$ = .126; Nagelkerke $r^2$ = .204; Global percentage 80.4.

the family could favor them for adequate weight gain during pregnancy. One study reported that single men (risk ratio [RR] 3.5, 95% confidence interval [CI] 4.2–9.4) and those divorced or separated (RR 6.3, 95% CI 9.5, 19.1) had a higher risk of dying from HIV infection than married men, although in women this was not the case [37], which demonstrates even further that in the case of women, having a partner does not necessarily reflect something that supports their health status.

Hasan et al reported in a study involving pregnant woman without HIV infection from Bangladesh that those with a lower socioeconomic status were at greater risk of inadequate weight gain in the third trimester of pregnancy [29], which is consistent with our findings of negative correlation between socioeconomic status and insufficient weight gain. However, we didn´t observe a correlation between schooling and insufficient weight gain [27, 29]. Moreover, in a study with HIV seropositive pregnant women without ART, lower weight gain was reported in women with a lower level of education as well as in those who did not contribute to household income [33].

In part, the contribution of socioeconomic factors to weight gain is through diet, this relationship has been studied more through types of food security and eating patterns [38] which, in turn, will be reflected in the nutrient intake. We found a positive correlation between the socioeconomic level and the consumption of energy and several nutrients (iron, zinc, protein, monounsaturated fat, and fiber), as well as a high prevalence of insufficient zinc and vitamins C intake in the lowest socioeconomic levels. This may represent an example of how socioeconomic level affects diet quality through eating patterns characteristic of vulnerable populations, such as low consumption of vegetables, fruits, seeds, and low-fat animal foods [38]. Similarly, a study involving Brazilian pregnant women concluded that per capita family income was one of the sociodemographic factors associated with adherence to a healthy diet (rich in legumes, fruits, and vegetables) during pregnancy [38]. On the other hand, in a study where behaviors were evaluated in terms of purchase of food, it was observed that food buyers residing in low-income households were less likely to buy foods high in fiber and low in fat, salt, and sugar [39]. Moreover, in the population with HIV infection, food insecurity is associated with poor adherence to ART and progression of the disease [7], so it is important to consider the socioeconomic level of seropositive pregnant women in strategies that seek to improve the quality of their diet for maintaining adequate weight.

In women who had CD4 count of <350, a greater proportion had inadequate intake of folic acid and zinc which may be due to a higher frequency of gastrointestinal symptoms as shown in Table 3, which may be related to not been on ART for long since it has been reported that gastrointestinal side effects are more frequent at the beginning of ART [40].

In this study, energy consumption was not related to weight gain unlike other studies involving pregnant women without HIV infection in which greater weight gain was found with higher energy consumption in the diet [28, 31]. However, there was a tendency to lose weight when the energy intake was insufficient in pregnant women with CD4 count of <350 (r = 0.3: p = 0.058). Another variable that was associated with poor weight gain in this study was the level of protein consumption in those with CD4 count <350, a result similar to that of a cohort study involving 224 pregnant women without HIV infection where the energy-adjusted intake of protein was associated with weight gain at the end of the second quarter [41].

The negative correlation observed between insufficient weight gain and multivitamin supplementation mainly in women with CD4 count of ≥350 shows the possible beneficial role of multivitamin supplementation in the maintenance of adequate weight gain in this population. Concerning the median weekly weight gain, Villamor [33], reported that HIV seropositive pregnant women not receiving ART but taking multivitamin were less likely to present low

rates of weekly weight gain. Consequently, it has been indicated that multivitamin supplementation is beneficial for HIV seropositive pregnant women without ART and their neonates and does not result in adverse effects, highlighting the need for more studies on pregnant women with HIV infection who are receiving ART [42].

About 52.7% of the women experienced at least one gastrointestinal symptom (GIS) due to ART which is lower than the rate reported in a cohort of 290 HIV-infected patients, where 74.5% of patients had at least one GIS [43]. The prevalence of nausea and vomiting in this sample was 42% and 28%, respectively, which is similar to that reported by Karus et al. in their multicenter study involving 376 HIV infected patients in which 42–57% and 23–28% experienced nausea and vomiting, respectively [44]. Although it has been reported that the side effects of ART are more frequent in women [43] and even more so in pregnant women [45], the prevalence of nausea and vomiting was similar to that reported in other studies in HIV infected patients which did not include pregnant women [43 and 44]. In the present study the presence of nausea and vomiting, referred as a result of the ART by the patient, affected the weight gain mainly in women with a count of <350 CD4 since they were most likely starting their ART, and they had more GIS. The presence of nausea (52.7%) and vomiting (50%) referred from pregnancy was similar to the ranges reported in pregnant women without HIV (50–80% for nausea and 50% for vomiting) [46]. Our study found suboptimal intake of zinc and folic acid. A related study found that pregnant women hospitalized with *hyperemesis gravidarum* showed lower energy and nutrients intake than pregnant women without this clinical condition [47].

Height is positively correlated with weight gain in pregnant women without HIV infection [30–32], as reflected in this study. In the case of pre-gestational BMI, we observed a negative correlation with insufficient weight gain, which is consistent with Popalet al. where pregnant overweight and obese pregnant women gained more weight [48]. Our findings were not consistent with Anderson et al. who found a lower weight gain in women with higher prepregnancy BMI [49].

In conclusion, factors related to insufficient weight gain in HIV seropositive pregnant women who use ART with a CD4 count of ≥350 include marital status, socioeconomic level, and supplementation with multivitamins. On the other hand, those who had CD4 count of <350 were at greater risk of having insufficient weight gain if they were in a consensual union, had gastrointestinal symptoms due to the use of ART (nausea and vomiting) and if their protein intake was less than 12%. Marital status was a factor that affected all women regardless of their immunological condition. Thus, in order to achieve an adequate weight gain in this population in addition to continuing ART to avoid vertical transmission and improve patients' immune status, personalized nutritional interventions are required. These should focus on ensuring an adequate intake of energy and nutrients that may as well as strategies to reduce gastrointestinal symptoms, provide economic support, and enhance the use of multivitamins and energy or protein supplementation.

One of the most remarkable findings in the present investigation, is that the marital status of pregnant women living with HIV infection is a significant factor that contributes to insufficient weight gain in pregnancy. Therefore, it is important to sensitize couples, educating them of the importance of contributing to promoting equitable family dynamics among the genders that favor healthy eating habits in HIV seropositive pregnant women. For this reason, we suggest the need to include the gender perspective [50,51] during discussions around weight gain of pregnant women in vulnerable conditions.

Although in this study, insufficient weight gain during pregnancy was defined by the socially vulnerable status of HIV seropositive women within the family environment as a result of gender inequality in food distribution and socioeconomic status, and not only because of

clinically controlled aspects such as diet or ARV consumption, we suggest the importance of a multidisciplinary and comprehensive approach including clinical interventions. In particular, it will be important to consider the socioeconomic situation and the gender perspective. This will most likely contribute to improving the nutritional status of pregnant women with HIV infection and their uninfected newborns exposed to HIV.

## Supporting information

**S1 Dataset.**
(XLSX)

## Author Contributions

**Conceptualization:** Estela Godínez, Mayra Chávez-Courtois.

**Data curation:** Estela Godínez.

**Formal analysis:** Estela Godínez, Mayra Chávez-Courtois, Ricardo Figueroa.

**Investigation:** Estela Godínez, Mayra Chávez-Courtois, Ricardo Figueroa, Rosa María Morales, Maricruz Tolentino.

**Methodology:** Estela Godínez, Mayra Chávez-Courtois, Ricardo Figueroa, Rosa María Morales, Cristina Ramírez, Maricruz Tolentino.

**Project administration:** Estela Godínez.

**Supervision:** Ricardo Figueroa.

**Writing – original draft:** Estela Godínez, Mayra Chávez-Courtois.

**Writing – review & editing:** Estela Godínez, Mayra Chávez-Courtois, Ricardo Figueroa, Rosa María Morales, Cristina Ramírez, Maricruz Tolentino.

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
