## [Decision Letter · Decision Letter 0]

8 Nov 2019

PONE-D-19-18897

INFLUENCING FACTORS CONCERNING INSUFFICIENT WEIGHT GAIN ON MEXICAN PREGNANT WOMEN WITH HIV AND ANTIRETROVIRAL THERAPY

PLOS ONE

Dear Dr Chávez-Courtois,

Thank you for submitting your manuscript to PLOS ONE. After careful consideration, we feel that it has merit but does not fully meet PLOS ONE’s publication criteria as it currently stands. Therefore, we invite you to submit a revised version of the manuscript that addresses the points raised during the review process.

The paper is important and contributes to the field especially because the study is conducted in an at risk population on a problem that is rarely studied (pregnancy weight gain with HIV infection) .  In addition to the reviewer comments, we would request the following to be addressed.

1.  There are many grammatical and typesetting errors.  Many sentences are fragments.  There are acronyms like ART that are not defined. The academic editor suggestions having the paper carefully proofread by a colleague who speaks English as their first language.   The academic editor has provided an attachment of handwritten notes, however, these are insufficient edits.  The entire manuscript needs to be edited.

2. Pregnancy weight gain may be incorrect because the study participants may not know their pre-pregnancy  weight.  A recent article in the journal Obesity, titled "Do women know their pre-pregnancy weight?" examined women from Mexico and there was evidence that pre-pregnancy weight may have not been known and the self-reported weight deviated from ground truth.  The academic editor suggests using the calculator from this paper to estimate pre-pregnancy weight from the first measured weight as a proxy to check.  Note that because the sample used in that study were healthy pregnancies, it may still be inaccurate.  This is okay, because the authors would have done everything possible to account for this.

We would appreciate receiving your revised manuscript by Dec 23 2019 11:59PM. To enhance the reproducibility of your results, we recommend that if applicable you deposit your laboratory protocols in protocols.io, where a protocol can be assigned its own identifier (DOI) such that it can be cited independently in the future. For instructions see: http://journals.plos.org/plosone/s/submission-guidelines#loc-laboratory-protocols

We look forward to receiving your revised manuscript.

Kind regards,

Diana M. Thomas

Academic Editor

PLOS ONE

Journal Requirements:

Reviewers' comments:

Reviewer's Responses to Questions

**Comments to the Author**

1. Is the manuscript technically sound, and do the data support the conclusions?

Reviewer #1: Yes

2. Has the statistical analysis been performed appropriately and rigorously? 

Reviewer #1: Yes

3. Have the authors made all data underlying the findings in their manuscript fully available?

Reviewer #1: Yes

4. Is the manuscript presented in an intelligible fashion and written in standard English?

Reviewer #1: No

5. Review Comments to the Author

Reviewer #1: See Attached. This is an important article and should be published. I suggest the employment of an academic editor, as the English presentation of the work is distracting from the results. Although this study focuses on the 112 women receiving ART, I suggest a similar study be accomplished on the general population of Mexican women, so the results of the women receiving ART be placed in appropriate academic context.

6. PLOS authors have the option to publish the peer review history of their article (what does this mean?). If published, this will include your full peer review and any attached files.

Reviewer #1: Yes: Jonathan W. Roginski, Ph.D.

---

## [Author Response · Author response to Decision Letter 0]

14 Jan 2020

Editor: Thanks about all your suggestions. We have incorporated all of your proposals into our revision. They were absolutely helpful. Thank you always. 

Reviewer: Thanks about all your suggestions. We have incorporated all of your proposals into our revision. They were absolutely helpful. Thank you always.

---

## [Editor Report · Decision Letter 1]

24 Jan 2020

PONE-D-19-18897R1

FACTORS ASSOCIATED WITH INSUFFICIENT WEIGHT GAIN AMONG MEXICAN PREGNANT WOMEN WITH HIV INFECTION RECEIVING ANTIRETROVIRAL THERAPY

PLOS ONE

Dear Dr Chávez-Courtois,

Thank you for submitting your manuscript to PLOS ONE. After careful consideration, we feel that it has merit but does not fully meet PLOS ONE’s publication criteria as it currently stands. Therefore, we invite you to submit a revised version of the manuscript that addresses the points raised during the review process.

The comments from Reviewer 1 were not responded to. There may have been changes made based off the comments, however, a point by point rebuttal as requested was not provided. Please provide this response.

We would appreciate receiving your revised manuscript by Mar 09 2020 11:59PM. To enhance the reproducibility of your results, we recommend that if applicable you deposit your laboratory protocols in protocols.io, where a protocol can be assigned its own identifier (DOI) such that it can be cited independently in the future. For instructions see: http://journals.plos.org/plosone/s/submission-guidelines#loc-laboratory-protocols

We look forward to receiving your revised manuscript.

Kind regards,

Diana M. Thomas

Academic Editor

PLOS ONE

Additional Editor Comments (if provided):

The comments from Reviewer 1 which was a PDF attachment need a point by point rebuttal. This was not included in the resubmission.

---

## [Author Response · Author response to Decision Letter 1]

12 Mar 2020

Editor: Thanks about all your suggestions. We have incorporated all of your proposals into our revision. They were absolutely helpful. Thank you always. 

Reviewer 1: Thanks about all your suggestions. We have incorporated all of your proposals into our revision. They were absolutely helpful. Thank you always.

---

## [Editor Report · Decision Letter 2]

17 Apr 2020

PONE-D-19-18897R2

FACTORS ASSOCIATED WITH INSUFFICIENT WEIGHT GAIN AMONG MEXICAN PREGNANT WOMEN WITH HIV INFECTION RECEIVING ANTIRETROVIRAL THERAPY

PLOS ONE

Dear Dr Chávez-Courtois,

Thank you for submitting your manuscript to PLOS ONE. After careful consideration, we feel that it has merit but does not fully meet PLOS ONE’s publication criteria as it currently stands. Therefore, we invite you to submit a revised version of the manuscript that addresses the points raised during the review process.

Please improve the English.

We would appreciate receiving your revised manuscript by Jun 01 2020 11:59PM. To enhance the reproducibility of your results, we recommend that if applicable you deposit your laboratory protocols in protocols.io, where a protocol can be assigned its own identifier (DOI) such that it can be cited independently in the future. For instructions see: http://journals.plos.org/plosone/s/submission-guidelines#loc-laboratory-protocols

We look forward to receiving your revised manuscript.

Kind regards,

Diana M. Thomas

Academic Editor

PLOS ONE

Additional Editor Comments (if provided):

Please revise by improving the English

---

## [Author Response · Author response to Decision Letter 2]

27 Apr 2020

Editor: Thanks about all your suggestions. We have incorporated all of your proposals into our revision. They were absolutely helpful. Thank you always. 

Reviewer: Thanks about all your suggestions. We have incorporated all of your proposals into our revision. They were absolutely helpful. Thank you always.

---

## [Editor Report · Decision Letter 3]

7 May 2020

FACTORS ASSOCIATED WITH INSUFFICIENT WEIGHT GAIN AMONG MEXICAN PREGNANT WOMEN WITH HIV INFECTION RECEIVING ANTIRETROVIRAL THERAPY

PONE-D-19-18897R3

Dear Dr. Chávez-Courtois,

We are pleased to inform you that your manuscript has been judged scientifically suitable for publication and will be formally accepted for publication once it complies with all outstanding technical requirements.

With kind regards,

Diana M. Thomas

Academic Editor

PLOS ONE

Additional Editor Comments:

It was a pleasure working with your team. Your findings were interesting and I enjoyed serving as your editor.

---

## [Editor Report · Acceptance letter]

12 May 2020

PONE-D-19-18897R3 

FACTORS ASSOCIATED WITH INSUFFICIENT WEIGHT GAIN AMONG MEXICAN PREGNANT WOMEN WITH HIV INFECTION RECEIVING ANTIRETROVIRAL THERAPY 

Dear Dr. Chávez-Courtois:

I am pleased to inform you that your manuscript has been deemed suitable for publication in PLOS ONE. Congratulations! Your manuscript is now with our production department. 

With kind regards,

on behalf of

Dr. Diana M. Thomas 

Academic Editor

PLOS ONE